# Why Did You Do That? Generalizing Causal Link Explanations to Fully Observable Non-Deterministic Planning Problems

**Sarath Sreedharan[1], Christian Muise[2] , Subbarao Kambhampati[1]**

[1] Arizona State University
[2]Queen's University

ssreedh3@asu.edu, christian.muise@queensu.ca, rao@asu.edu

## Abstract

The problem of designing automated agents, particularly automated planning agents that can explain their decisions has been receiving a lot of attention in recent years. The field of explainable planning or XAIP has already made a lot of progress in recent years and many of them centered around the problem of explaining decisions derived for classical planning problems. As the field progresses there is interest in tackling problems from more complex planning formalisms. However, one important aspect to keep in mind as we start focusing on such settings is that the explanatory challenges we study in the context of classical planning problems do not disappear when we move to more general settings but are just magnified. As such, when we move to these more general settings, a significant challenge before us is to see how one could generalize the well-established methods studied in the context of classical planning problems to these new settings. To provide a concrete example for this new research program we will start with causal link explanations, one of the earliest and most widely used explanations techniques used in the context of policies generated for fully observable non-deterministic planning problems. This would see us generalizing a concept that was originally developed for a specific solution concept, i.e, sequential plans, and see them applied to a very different solution concept (rooted policies). We will develop a compilation-based method for generating generalized causal link explanations and show how as the domain is limited to deterministic cases, our method would generate causal link chains as identified by earlier works.

## 1 Introduction

As AI systems tackle more and more complex problems, the need to explain their reasoning in intuitive terms to their users becomes ever more pressing. This has led to a lot of interest in studying and building explainable AI systems [Gunning and Aha, 2019]. In particular, AI planning has been getting a lot of attention as an ideal testbed for developing and testing explainable AI techniques (XAI) [Fox *et al.*, 2017].

This interest stems both from the complexity of problems modern planners can handle and the availability of human-readable symbolic models that are used by these planners. There have also been recent cases made for using learned post hoc symbolic models to provide explanations to sequential decision-making problems, even when the original problem may not be framed in those terms [Kambhampati *et al.*, 2012; Sreedharan and Kambhampati, 2021]. Despite all the interest in explainable planning, the vast majority of the recent works in explainable planning (XAIP) focus on deterministic fully observable planning problems (referred to popularly as classical planning problems) [Chakraborti *et al.*, 2020]. In many ways this is not surprising, after all as the name suggests, classical planning is indeed one of the most mature subfields within automated planning [Geffner and Bonet, 2013]. However, the reasons for the interest in this particular style of planning problems go beyond just the availability of scalable planners. Arguably one of the big draws of classical planning problems as an ideal research platform for XAIP is that it presents a clean formulation of sequential decision-making problems without other complexities. This allows one to focus just on addressing some of the central explanatory problems related to sequential decision-making without worrying about all the challenges raised by the various relaxations made by each planning formulation.

As both the field of XAIP and planning grows we can no longer just focus our attention on classical planning problems and start considering more expressive planning formalisms. In this particular paper, we would like to take up the case of generating explanations in the context of FOND or Fully Observable Non-Deterministic planning problem. In recent years FOND planners have been applied in various application domains including dialogue generation [Muise *et al.*, 2019] and have also been used as the basis for deriving generalized plans [Bonet *et al.*, 2019]. Though the utility of explanation in FOND goes beyond just explaining decisions derived by native FOND planners. By providing a qualitative modeling framework for non-determinism, FOND planners with appropriate assumptions about fairness could also be used to provide post hoc explanations for decisions derived by stochastic planners. It is widely known that people are poor probabilistic reasoners [Tversky and Kahneman, 1993], as such for most interactions with everyday users the system could provide more intuitive explanations by first mapping

the decision-making problem into a FOND problem.

However, the goal of this paper is not just to introduce a set of new techniques for generating explanations for FOND problems but to also push for a new research agenda. One that recognizes that the explanatory problems studied in the context of classical planning problems do not disappear once we move to more complex decision-making settings, but are only magnified by the introduction of the other complexities. As such we should treat the work done in generating explanations for classical planning problems as a starting point to develop more general methods that apply to the problems at hand. As a demonstration of this research philosophy, we will start with the causal chain explanations [Seegebarth *et al.*, 2012]. Such explanations are one of the earliest and most widely used explanation strategies that have been considered in the context of classical planning problems. We will propose a more general version of it that can be applied in the context of FOND problems. This would involve us mapping the concept of causal justification of action in a plan step that was developed in the context of sequential plans and mapping them over to a separate solution concept that is more common in the context of FOND namely policies that map states to actions. We will present a compilation-based method for generating generalized causal chain explanations and show how the method generates causal chains of the form discussed in the context of classical planning problems when the domain is limited to deterministic domains. The paper also evaluates the effectiveness of the proposed method by analyzing the computational characteristics of the method over standard FOND benchmarks.

## 2 Background

We will be focusing on cases where the planning problem may be represented as a fully observable non-deterministic planning problem (FOND). Such models may be represented in declarative form using PDDL variants that use 'oneof' effects [Bryce and Buffet, 2008]. Mathematically, we expect a FOND model to be represented by a tuple of the form $\mathcal{M} = \langle F, A, I, G \rangle$, where, $F$ is a set of propositional fluents that is used to define the state space for the planning problem ($S = 2^F$); $A$ is the set of actions available to the agent; $I \subseteq F$ is the initial state from which the agent needs to try achieving the goal; and $G \subseteq F$ is the goal specification and any state that satisfy the goal specification (i.e., $G \subseteq s$) is considered to be a valid goal state. To simplify the discussion, without loss of generality, we will assume $G$ is a singleton set consisting of a single goal atom. Overloading the notation a bit, we will also use the symbol $G$ to denote the goal atom. Each action $a \in A$ is further defied by a tuple $a = \langle pre_a, \mathbb{E}(a) \rangle$. In this action definition, $pre_a$ stands for the preconditions for executing the action and $\mathbb{E}(a)$ the set of possible effects. In this paper, we will exclusively focus on positive conjunctive preconditions, as such we will represent each precondition as a subset of fluents. $\mathbb{E}(a) = \{\langle add_a^1, del_a^1 \rangle, ...., \langle add_a^k, del_a^k \rangle\}$ represents the set of mutually exclusive effects that could occur as the result of executing the action $a$ and $add_a^i \subseteq F$ and $del_a^i \subseteq F$ correspond to the add and delete effect corresponding to the $i^{th}$ effect. With the action definitions in place we

can also define the set of transitions possible under this action definition, denoted as $\mathbb{T}$, such that we define a transition $\langle s_1, a, s_2 \rangle$ to be possible (denoted as $\langle s_1, a, s_2 \rangle \in \mathbb{T}$) if

$$s_1 \subseteq pre_a \text{ and } \exists j, \text{ such that}$$
$$\langle add_a^j, del_a^j \rangle \in \mathbb{E}(a), \ s_2 = (s_1 \setminus del_a^j) \cup add_a^j$$

Throughout this paper, we will focus on cases where non-determinism is considered to be fair, i.e., for every non-deterministic action every possible effect is guaranteed to occur if the action is executed infinitely often. A solution to a FOND problem takes the form of a policy that maps a state to action, usually denoted by a function $\pi : S \to A \cup \{a^\emptyset\}$), where $a^\emptyset$ is an artificial empty action assigned to states that are either not supported by the policy or are goal states. A concept that will be central to the main of the techniques are traces supported by a given policy. We will refer to a state action state sequence of the form $\tau = \langle s_1, a_1, ..., s_k \rangle$ as a trace supported by a policy $\pi$ if for every $s_i$, where $i \neq k$, we have $\pi(s_i) = a_i$, $\langle s_i, a_i, s_{i+1} \rangle \in \mathbb{T}$. A trace is said to be a goal achieving trace if $G \subseteq s_k$ and a state state $s_j$ is said to be reachable from $s_i$ if there exists a trace of the form $\tau = \langle s_i, a_i, ..., s_j \rangle$.

In terms of a valid policy for a FOND problem, the literature generally differentiates between weak solutions, strong and strong-cyclic solutions. Weak solutions are policies such that there exists at least one goal-achieving trace from the initial state. A policy is said to be strong-cyclic if the goal is reachable from all states reachable from the initial state. Finally, a policy is said to be a strong solution if we can again guarantee that goal is reachable from all states reachable from the initial state, but additionally, now we require that a state can never be repeated in any given goal-reaching trace. However, in this paper we will not differentiate between these specific classes of solutions and all methods studied here are equally valid for all classes of valid policies.

**Causal Chain Explanations** As discussed earlier, in this paper we will primarily focus on leveraging intuitions from and generalizing a specific explanation technique studied in the context of classical planning problems called causal link chain explanations. As per [Seegebarth *et al.*, 2012], the exact explanatory query being addressed here is

"Why is the step 'o:a "necessary" for $\pi$ to constitute a solution?"

Where each plan step consists of a label (denoted by 'o' in the query) and an action (denoted by 'a'), and the explanation takes the form of a sequence of causal links that originates at the step in question and terminates at the goal. Now a causal link is said to exist between two-step $o_i : a_i$ and $o_j : a_j$ if there exists a precondition for the action $a_j$ that is provided by the add effect of $a_i$. The causal link between the action is denoted as $o_i : a_i \to_p o_j : a_j$, where $p$ is the fact being 'produced' by action $a_i$ and being consumed by $a_j$. Each causal link is assumed to be not threatened by any other action, and no action between the producer and consumer actions could have supported that precondition. The goal of causal chain explanation is to establish that an action is justified because it helps establish some fact for another action, however, as

[Fink and Yang, 1993] points out, there could be different notions of justification in this context. If we focus merely on an action establishing a precondition, while no intermediate action threatens the causal link or adds the fact, then it corresponds to the category called *Backward Justified* actions. However, the removal of a backward justified action doesn't necessarily result in a solution being invalid (as the facts may have been already added by a previous action). This brings us to the notion of *Well-Justified* actions. An action is said to be well-justified if the removal of that action will cause the resultant plan to be invalid. In this paper, we will focus exclusively on explanations that establish that the action is well justified, and require that each causal link representing a precondition establishment could not have been established by previous actions. A plan is said to be well-justified if every action in the plan is well-justified. It is worth noting however that this is not the strongest notion of justification, in fact, one could talk about *Perfectly Justified* plans. Specifically, a plan is said to be perfectly justified if there exists no subset of actions that can be removed, while preserving the validity of the plan, i.e., the goal is still achievable by the remaining sequence. However, in this paper, we won't delve any further into establishing or explaining perfectly justified plans. Justification can be used as a basis for a weaker form of optimality, one that argues that each action in the plan serves a purpose, in fact, a perfectly justified plan is referred to as a minimal plan [Kambhampati, 1995].

## 3   Motivating Example

As a running example throughout the paper consider the policy generated by a futuristic daily planner that takes into account all the possible contingencies of the day and comes up with a policy that will get you to the office in time. The policy starts with you at home and as the first action, the daily planner recommends you to start the day by placing a call to your local baker for a dozen of the day's special donuts. At the end of this action, you will find yourself at home having ordered a dozen of maple-glazed donuts or a dozen strawberry sprinkle donuts with a coupon for a free milkshake. Now based on the outcome of this action the policy now requires you to take different routes to the office, with different potential branching points owing to the various non-determinisms in the world. Figure 1, presents a high-level overview of this policy with its various contingencies. Regardless of your personal feelings toward fried pastry, you may be confused as to why your daily planner might be asking you to take the time to order and pick up donuts when you should be trying to get as early as possible to the office parking lot to get a free parking spot. Looking at the immediate actions that follow, one may be forgiven to think that the action is just a random action thrown into a seemingly pointless plan created by a faulty planner. Additionally, you may not have the patience to go through each possible trace corresponding to the multitude of ways the world may evolve and how they may feed into your goal of getting to your office. Ideally, you would want to be able to leverage mechanisms like causal chain explanations that demonstrate why the action is required for you to get to the goal. However, as we move to the non-deterministic setting,

we don't even have a clear notion of when action may be required for a policy to be valid. So in this paper, we will start by providing a formalization of when an action may be required for the achievement of the goal and provide an explanation strategy that will allow us to explain this fact to a user of the system. Additionally, we will see that this explanation strategy is a genuine generalization of the notion of causal link explanations from classical planning settings.

## 4   Explaining Action Requirement

The specific explanatory query we are interested in studying is the question

*"Why is the action a required at state s in policy $\pi$ for a planning problem $\mathcal{M}$?"*

Which is just a mapping of the question studied in [Seegebarth *et al.*, 2012] to the FOND setting and we will use the notation $\langle s, a, \pi, \mathcal{M} \rangle$ to denote the specific query. The first order of business here is to quantify exactly when an action is required for a goal. We will say that an action is required for the policy if it is well-justified. Repeating the definition in the context of sequential plans, one could informally say that an action 'a' is well-justified at state 's' if without the execution of action 'a' at state 's' the goal could not have been achieved by the rest of the policy. However, this is not an operationalizable description of the property as by the very nature of policy as a solution concept, the execution of an action is necessary as the change of the state is needed to enable the execution of the rest of the policy. While in the context of sequential plans, one could meaningfully talk about removing an action from the sequence and then testing whether the remaining plan is valid or not, it is unclear how one could perform such transformations over a policy. At the same time, it is worth remembering that the concept of whether or not an action is well-justified at a particular policy step is still a relevant question to ask. After all, it would make no sense to claim that one could make a non-well-justified sequential plan well-justified by just mapping it to a policy. In this paper, we will try to propose a formal definition of this concept that leverages the fact that from any given state, one could characterize how the policy contributes to the goal by considering all the goal-achieving traces.

**Definition 1.** *An action 'a' is said to be* **required** *(or equivalently* **well-justified**) *at a state 's' for a policy '$\pi$' to achieve a goal G, if for every goal-reaching trace originating at state 's' the action 'a' is well-justified.*

In our example discussed above, that means that every outcome of the action 'order_donuts' contributes at least one useful fact that may be needed by some future actions.

Note that the notion of an action being required is an extremely strong condition, and there could very well be goal-reaching policies where none of the actions are required (a fact that is true for "well-justified" actions in classical planning as well). One could also look at weaker notions of how an action contributes to a goal (for example if the action 'a' is well-justified in at least one of the trace or 'a' may be well-justified for some subset of traces), however, that also means that one could in principle build a valid weak solution with

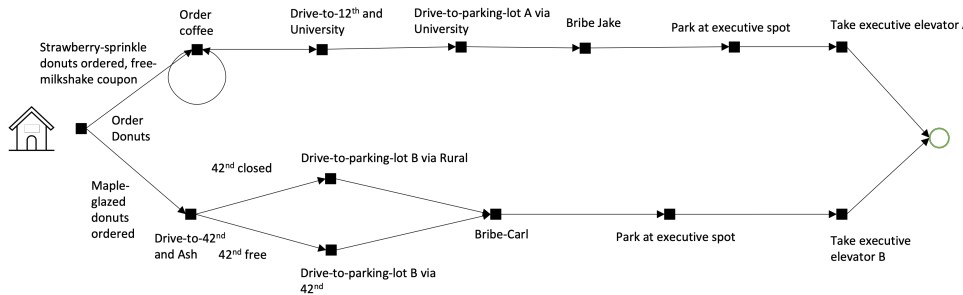

Figure 1: A simple overview of the daily planner policy, which highlights the actions that are determined by the policy along with some of the non-deterministic effects caused by the action.

the rest of the policy while ignoring the current action. We will leave the investigation of such weaker forms of justifications and how they may correspond to existing notions of justification for sequential plans as future work. Additionally, we will also leave the question of extending these notions to the policy as a whole as future work.

Now with the central property that we hope to explain in place, the next question is how to explain this property to a user? One obvious strategy could be to just enumerate all possible goal-reaching traces and present a causal link contributed by the action. However, even in the smallest domains, this explanatory scheme would overwhelm most users. Instead, we will focus on generating a more abstract explanation that will leverage necessary subgoals made feasible by the execution of the action and the explanation takes the form of a chain of such subgoals, where the achievement of the first subgoal in the sequence requires the execution of the action and any subsequent subgoals require the achievement of earlier subgoals. In particular, for grounding the concept of a necessary subgoal we will build on the notion of a policy landmark introduced in [Sreedharan et al., 2020b], which defined policy landmarks as being facts and their corresponding ordering that needs to be satisfied by every trace with a non-zero probability that can be sampled from the initial state. In our case, we will use a more restricted version of policy landmarks one that additionally requires that the landmarks we focus on are required as preconditions for different actions. We will refer to such policy landmarks as *causal policy landmarks*. By focusing on causal policy landmarks, we effectively filter out any facts that just appear as side-effects of some actions and only focus on the facts used by actions in the policy. In the case of our daily planner domain, a possible subgoal sequence could be

*We need to perform* `order_donuts` *to achieve the subgoal sequence*

```
security-guard-bribed →
parked-at-executive-parking-spot→
in-executive-elevator → at-office.
```

## 4.1 Generating Subgoal Sequence

As in the case of [Sreedharan et al., 2020b], we will be leveraging the all outcome determinization [Yoon et al., 2007] of the problem to identify the landmarks, but we will be leveraging a new compilation to be able to separate the landmarks

that are part of action preconditions from the ones that may just be side effects.

In particular, we will be leveraging a formulation that maintains two copies of each fluent, i.e, for each fluent $f$ that is part of the original problem definition we will introduce a new fluent $f^\phi$. We will maintain the mapping between the two copies using the function $\phi : f \mapsto f^\phi$ and also overload the function to also apply to sets.

For a given model $\mathcal{M} = \langle F, A, I, G \rangle$ and a query regarding the use of action $a$ in state $s$ for policy $\pi$, we will be creating a new model that will allow for the use of such duplicate fluents, i.e., $\mathcal{M}^\pi_{\langle s,a \rangle} = \langle F^\pi_{\langle s,a \rangle}, A^\pi_{\langle s,a \rangle}, I^\pi_{\langle s,a \rangle}, G^\pi_{\langle s,a \rangle} \rangle$, such that

$F^\pi_{\langle s,a \rangle} = F \cup \phi(F)$
$A^\pi_{\langle s,a \rangle} = \{a_i^{s_j} \mid \langle s_j, a_i \rangle \in \pi\}$
$I^\pi_{\langle s,a \rangle} = s \cup \phi(s)$
$G^\pi_{\langle s,a \rangle} = G$

Note that the new initial state corresponds to the state $s$ part of the query and contains fluents from both the original fluent set and the new copy fluent set. Each new action in the set $A^\pi_{\langle s,a \rangle}$ correspond to a specific state action mapping defined in the policy $\pi$. Specifically, an action $a_i^{s_j} \in A^\pi_{\langle s,a \rangle}$ will encode the fact that this copy of the action $a_i$ is meant to be executed only in state $s_j$, but also has preconditions that may only be a subset of the fluents that are true in $s_j$. The former captured in terms of the fluents belonging to the set $\phi F$ and the latter by using fluents from the original set $F$. Similarly, action effects will now include copies of the original effects of the action in terms of both fluent set, thereby allowing us to capture both the actions capability of allowing the continued execution of the policy while allowing us to maintain a separate accounting of how the action contribute to the preconditions of future actions. More formally, the action will be defined as $a_i^{s_j} = \langle pre_{a_i^{s_j}}, \mathbb{E}(a_i^{s_j}) \rangle$, such that

$pre_{a_i^{s_j}} = \phi(s_j) \cup pre_{a_i}$
$\mathbb{E}(a_i^{s_j}) = \{\langle \phi(add_{a_i}^m) \cup add_{a_i}^m, \phi(del_{a_i}^m) \cup del_{a_i}^m \rangle \mid \langle add_{a_i}^m, del_{a_i}^m \rangle \in \mathbb{E}(a_i)\}$

One point to note here that is that to effectively constrain application of actions to specific states in the policy, we have to not only consider facts that are true in the state but also the ones that are false. We can still use our positive precondition formulation to support this by using the standard compilation technique to compile away negative preconditions. Since this

is a standard technique, we will not include this as part of our formalization, but the reader is advised to keep in mind that when we say $\phi(s)$ is part of the precondition it also includes new positive fluents that corresponds to the fluents that may be false in state $s$ (with the necessary changes made to the effects as well).

Now the resultant model $\mathcal{M}^\pi_{\langle s,a \rangle}$ is still a non-deterministic planning domain. To generate landmarks, we will be considering an all outcome determinization of the model $\mathcal{D}(\mathcal{M}^\pi_{\langle s,a \rangle})$. Given the nature of the determinization, the set of goal-reaching traces for the model $\mathcal{M}^\pi_{\langle s,a \rangle}$ would exactly correspond to the set of valid plans for the deterministic model $\mathcal{D}(\mathcal{M}^\pi_{\langle s,a \rangle})$. This brings us to the first proposition

Now, we will be using the $\mathcal{D}(\mathcal{M}^\pi_{\langle s,a \rangle})$ to identify the causal landmark set [Keyder *et al.*, 2010], where causal landmarks are landmarks that correspond to landmarks that always appear in the precondition of an action. However, for our purposes, we can't directly use the causal landmark set generated from $\mathcal{D}(\mathcal{M}^\pi_{\langle s,a \rangle})$ as the preconditions of the actions in the model also contain state descriptions. As such the landmarks directly calculated from $\mathcal{D}(\mathcal{M}^\pi_{\langle s,a \rangle})$ may contain facts that are not part of any action preconditions. Our use of a distinct set of fluents to capture the state and preconditions allows us to filter out such landmarks. Specifically, let $\mathbb{L} = \langle \mathcal{L}, \prec \rangle$ be the landmark set, where $\mathcal{L} \subseteq F^\pi_{\langle s,a \rangle}$ is the set of landmark fluents and $\prec$ is the ordering between the fluents (we will specifically focus on sound ordering derived from delete relaxations of the problem [Richter *et al.*, 2008]), then we will use the set $\mathbb{L}' = \langle \mathcal{L}', \prec \rangle$, where $\mathcal{L}' = \mathcal{L} \setminus \phi(F)$.

**Proposition 1.** *The landmark set $\mathbb{L}'$ for the model $\mathcal{D}(\mathcal{M}^\pi_{\langle s,a \rangle})$ correspond to the causal policy landmarks for policy $\pi$.*

We can establish this proof by following a slightly modified version of the proof described in [Sreedharan *et al.*, 2020b]. It's also worth noting that, we are guaranteed that $\mathcal{G} \subseteq \mathcal{L}'$

Before we can generate our explanation, we need to identify the landmarks whose achievement actually requires the execution of the action in question ($a$) at state $s$. To identify whether a landmark $f \in \mathcal{L}$ requires the action, we will be using a modified version of the model $\mathcal{M}^\pi_{\langle s,a \rangle}$, denoted as $\hat{\mathcal{M}}^\pi_{\langle s,a \rangle, \to f} = \langle F^\pi_{\langle s,a \rangle}, \hat{A}^\pi_{\langle s,a \rangle}, I^\pi_{\langle s,a \rangle}, \{f\} \rangle$ The first thing to note is that the goal of the new problem is to achieve $\{f\}$. The next change is that of introduction of the new action set $\hat{A}^\pi_{\langle s,a \rangle}$. In particular $\hat{A}^\pi_{\langle s,a \rangle}$ is formed from $A^\pi_{\langle s,a \rangle}$ by removing the action corresponding to the query state action pair ($a^s$) and replacing it with a new action ($\hat{a}^s$) that will allow for the policy execution but will not contribute to preconditions of any future action. Specifically, we will define the action as
$pre_{\hat{a}^s} = \phi(s) \cup pre_a$
$\mathbb{E}(\hat{a}^s) = \{\langle \phi(add^m_{a_i}), \phi(del^m_{a_i}) \rangle \mid \langle add^m_{a_i}, del^m_{a_i} \rangle \in \mathbb{E}(a_i)\}$
Now we can use the formulation to identify whether the action was required by testing the solvability of this modified problem. In particular, we will have

**Proposition 2.** *An action 'a' is required at a state 's' for a policy '$\pi$' to achieve a landmark $f$ (where requirement is defined as per Definition 1), if and only if the modified planning*

problem $\mathcal{D}(\hat{\mathcal{M}}^\pi_{\langle s,a \rangle, \to f})$ *is unsolvable.*

*Proof Sketch.* To show the validity of the 'if' part, we first need to remember that $f$ was a policy landmark, and thus every trace from $s$ at one point led to a state containing $f$. Thus if there was a path whose validity didn't depend on a causal link contributed by the action 'a', that path should remain still valid under the modified model $\hat{\mathcal{M}}^\pi_{\langle s,a \rangle, \to f}$. The 'only if' part can be shown by a symmetric argument. $\square$

Finally, we can establish that the action was never required if we show that the compilation for the goal $G$ (i.e. $\mathcal{D}(\hat{\mathcal{M}}^\pi_{\langle s,a \rangle, \to G})$) is unsolvable.

However to provide the explanation we have to not only identify a single landmark that is required, but ideally, we would like to present a chain of facts each requiring the last fact to be achieved. Note that here we can't just rely on the landmark ordering as it may also encode the relationship being enforced by the state part of the preconditions. So we will build a variation of $\mathcal{M}^\pi_{\langle s,a \rangle}$ denoted as $\hat{\mathcal{M}}^\pi_{\langle s,a \rangle, f_1 \to f_2}$ that will try to identify such requirement relationship between landmarks. Specifically, we will have $\hat{\mathcal{M}}^\pi_{\langle s,a \rangle, f_1 \to f_2} = \langle F^\pi_{\langle s,a \rangle}, \hat{A}^\pi_{\langle s,a \rangle, f_1 \to f_2}, I^\pi_{\langle s,a \rangle}, \{f_2\} \rangle$. Now the goal is to achieve $f_2$, and we will form $\hat{A}^\pi_{\langle s,a \rangle, f_1 \to f_2}$ from $A^\pi_{\langle s,a \rangle}$ by removing $f_1$ from all add effects while preserving $\phi(f_1)$. More formally, let $a^{s'}_j \in A^\pi_{\langle s,a \rangle}$, then we have a correspond action $\hat{a}^{s'}_{j, f_1 \to f_2} \in \hat{A}^\pi_{\langle s,a \rangle, f_1 \to f_2}$, such that
$\mathbb{E}(\hat{a}^{s'}_{j, f_1 \to f_2}) = \{\langle add^{s',m}_{a_j} \setminus f_1, del^{s',m}_{a_j} \setminus f_1 \rangle \mid \langle add^{s',m}_{a_j}, del^{s',m}_{a_j} \rangle \in \mathbb{E}(a^{s'}_j)\}$

Since the requirement ordering will be a subset of the landmark ordering, we will only need to run this test between landmarks when there already exists an ordering. We will denote this requirement ordering with the notation $\prec_R$.

Finally, to generate the explanation chain itself, we will iterate over a topological sort over $\mathcal{L}'$ and find the first landmark $f_1$ that requires action $a$ and build a chain consisting of a set of totally ordered landmarks over the requirement ordering that terminates with the goal $G$. More formally

**Definition 2.** *A chain of facts $\mathcal{E} = \langle f_1, ..., f_j, ..., f_n \rangle$, such that all $f_i \in F$ is considered a **valid explanation** for the query $\langle s, a, \pi, \mathcal{M} \rangle$, if*

1. *Every fact $f_i$ in $\mathcal{E}$ is a causal policy landmark for the policy $\pi$ and model $\mathcal{M}$*

2. *$f_1$ requires the action 'a' to be executed in state $s$*

3. *For all pairs of landmarks, $f_i$ and $f_{i+1}$, we have $f_i \prec_R f_{i+1}$*

4. *Finally, we have $f_n = G$.*

The above definition presents a general description for a valid explanation. Note that the set of valid explanation covered by the above definition may not be equivalent in how effective the user may find them to be. As such one may need to use additional criteria to choose an explanation from this set of valid explanations. Choosing a landmark with the least

number of preceding facts as the first element in the chain being one such possible criteria.

## 4.2 Relationship to Causal Link Explanations

Now to see how these explanations compare against the causal chains, we will constrain ourselves to deterministic settings, where every action has a single outcome. Thus from any state, there can at most be one goal-achieving trace. We will assume the same policy structure. Now we will show that every valid explanation (per Definition 2) corresponds to the fact that are part of a causal chain explanation and every causal chain explanation correspond to an explanation of the form described in Definition 2.

**Proposition 3.** *For a given causal chain explanation $\langle s_1 : a_1 \rightarrow_{f_1} s_2 : a_2, ...., s_m : a_m \rightarrow_g s_g : a^\emptyset \rangle$, the chain $\mathcal{E} = \langle f_1, ..., g \rangle$ is a valid explanation for the requirement query $\langle s_1, a_1, \pi, \mathcal{M} \rangle$, when $\mathcal{M}$ is completely deterministic.*

*Proof Sketch.* To see why this is true, we can see that all three requirements of a valid explanation provided in Definition 2 are met here. (1.) directly holds as all the facts are causal policy landmarks (they all appear in the precondition and there is only one path). (2.) holds automatically as this is a fact that is contributed by the action and per our definition of causal link explanation no action between the producer and consumer would generate the fact $f_1$. Thus $f_1$ would cause $\hat{\mathcal{M}}^\pi_{\langle s,a \rangle, \rightarrow f_1}$ to be unsolvable as model will disallow any use of actions after $s_2$ to be used. (3.) holds because the causal links are preconditions and as such removal of them causes the subsequent fact to be unachievable at the subsequent step. □

Similarly, we can also show that

**Proposition 4.** *For any valid explanation chain $\mathcal{E} = \langle f_1, ..., g \rangle$ for the query $\langle s_1, a_1, \pi, \mathcal{M} \rangle$ (where $\mathcal{M}$ is completely deterministic), there exist a causal chain explanation of the form $\langle s_1 : a_1 \rightarrow_{f_1} s_2 : a_2, ...., s_m : a_m \rightarrow_g s_g : a^\emptyset \rangle$, for some action set $\{a_2, ..., a_m\}$.*

The proof for this proposition follows a similar line of argument to the one described in Proposition 3.

## 5 Empirical Evaluation

As a way to provide a preliminary evaluation of the explanation generation methods discussed in this paper, we ran our method on several standard FOND benchmarks [Muise, 2018]. In the evaluation, we were interested in identifying (a) the frequency with which well-justified action occurs in policies generated for these planners (b) time-taken to generate the explanation chain, and (c) the average length of the explanation chains generated. For generating the policies, we used the PRP planner [Muise *et al.*, 2012] which by default produces a policy defined over partial states. We generate the full state policy by executing this policy defined over partial states from the initial state (favoring actions with lower distance when multiple partial states match).

For generating the landmarks, we made use of the implementation of [Keyder *et al.*, 2010] provided by the FastDownward system [Helmert, 2006]. Additionally, we again used the FastDownward planner to test the unsolvability of the various subgoals. Table 1, provides an overview of the various statistics we calculated from the experiments. The experiments were run on four domains and five problems instance per domain (the first five provided in the benchmark). We skipped one for exploding blocks world and one for the zeno-travel as the goal conditions were already true in the initial state. For each policy, you can note that a significant number of non-trivial action pairs are well justified. By non-trivial state-action pairs, we refer to any reachable state action pair where the action didn't correspond to the goal. The generation time for the causal chains were quite within the limits to be applicable for systems that require quick response time. In fact for all domains except Triangle-tireworld the average time taken for explanations generated was less than four seconds. The maximum average chain length we obtained was three. Note that, currently for the causal chain generation, we were merely trying to find a chain to the goal and were not trying to calculate the shortest or the longest chain.

## 6 Related Work

The history of causal chain explanation starts much earlier than their latest incarnation in [Seegebarth *et al.*, 2012]. One of the earliest works to look at a similar form of information was the PRIAR system [Kambhampati, 1991] that introduced the notion of validation structures that encodes such information in the form of plan annotations. Validation structures were proposed as a correctness explanation that could then be used to guide various tasks including plan retrieval, refitting, and modification [Kambhampati, 1990]. Another early work that looked at the introduction of similar information was that of [Veloso, 1992], which looked at performing regression-based analyses to determine initial state conditions relevant to the goal. In more recent work, such information was also used by [Chakraborti *et al.*, 2019] to provide an overview of the plan as a whole. [Bryce *et al.*, 2017] also looks at similar information to visualize plans by visualizing causal link chains in the style of metro rail maps. [Bercher *et al.*, 2014] presents human subject studies to verify the effectiveness of such explanations by grounding these explanations in the context of the application of an assistive system for putting together a home theater system.

In terms of existing works that have tried to extend approaches and methods that were developed in the context of classical planning problems to other planning formalisms, most of the works tend to focus on methods that were already not tightly connected to the specific planning formalisms which were used by the original paper that introduced it. Some popular examples include the use of model-reconciliation techniques [Sreedharan *et al.*, 2021a] and the use of abstraction to explain unsolvability or plan infeasibility [Sreedharan *et al.*, 2021b; Sreedharan *et al.*, 2019b]. Model-reconciliation is easy to extend to other formulations as one could technically apply the same principle of reconciliation to any other model formulation. To the best of our knowledge, the principle of model reconciliation has been applied to MDPs [Sreedharan *et al.*, 2019a], propositional knowledge bases [Vasileiou *et al.*, 2020] and numeric planning models

| Domains | Problems | Policy Type | % of Non-Trivial Well-Justified State-action Pairs | Average Chain Length | Average Time Taken (Secs) |
|---|---|---|---|---|---|
| Exploding Blocksworld | prob 1 | Strong Cyclic | 100% | 2.71 | 2.97 |
| | prob 2 | Strong Cyclic | 100% | 3 | 3.18 |
| | prob 3 | Strong Cyclic | 100% | 2.52 | 2.85 |
| | prob 4 | Weak Cyclic | 75% | 2.42 | 2.80 |
| Tireworld | prob 1 | Weak Cyclic | 100% | 2.8 | 3.34 |
| | prob 2 | Strong Cyclic | 100% | 2 | 2.21 |
| | prob 3 | Strong Cyclic | 66.66% | 2.5 | 2.87 |
| | prob 4 | Strong Cyclic | 100% | 3 | 3.36 |
| | prob 5 | Strong Cyclic | 100% | 2.5 | 3.03 |
| Triangle Tireworld | prob 1 | Strong Cyclic | 100% | 3 | 3.28 |
| | prob 2 | Strong Cyclic | 100% | 2.5 | 3.17 |
| | prob 3 | Strong Cyclic | 66.66% | 2.33 | 5.34 |
| | prob 4 | Strong Cyclic | 100% | 2.25 | 12.39 |
| | prob 5 | Strong Cyclic | 100% | 2.2 | 30.36 |
| Zenotravel | prob 2 | Strong Cyclic | 100% | 2.57 | 2.99 |
| | prob 3 | Strong Cyclic | 95% | 2.4 | 2.76 |
| | prob 4 | Strong Cyclic | 100% | 2.83 | 3.23 |
| | prob 5 | Strong Cyclic | 100% | 2.57 | 2.91 |

Table 1: The evaluation of the proposed method on standard FOND benchmarks. Here non-trivial state action pair refers to state action pairs where the policy didn't assign the $a^{\emptyset}$ goal action.

[Vasileiou *et al.*, ]. The thread of using abstraction to explain unsolvability in the context of FOND problems [Sreedharan *et al.*, 2020a]. To the best of our knowledge, this is the only other explanation work that has looked at FOND problems.

## 7 Conclusion and Discussion

The paper presents a generalization of causal chain explanations to novel settings. In particular, we looked at how we can use such explanations to justify why an action may be required at a particular state and proposed a compilation based method to generate such explanation. We additionally saw some of the computational characteristics of the discussed method. However, even in the context of such generalized causal chain explanation for FOND problems, there are multiple next steps to be considered. For one, we need to run user studies to identify how helpful these explanations are. Among all the possible causal chains we could generate for a given query, people may have specific preferences on what they would perceive as the most helpful explanation. Additionally there is the question what additional information we could provide along with these chains that may further help the user understand the role the action may play in the policy. Possible information, here could include providing an exemplary trace from the current state to the goal along with the causal chain or providing information about future actions that may use these facts as preconditions, etc.

Then there is the various possible weaker notions of how an action might contribute to the achievement of the goal. As discussed there may be cases where an action may not be required to get to the goal, though it may still be helping the policy achieve the goal in certain traces. The question of how

we can detect such cases and provide effective explanations in such cases is still an open question.

Finally, we hope that more work would consider other problems of generalizing explanation methods designed for classical planning problems to other planning formalisms. Some obvious next steps include generalizing causal link to other planning formalisms (temporal plans, stochastic plans) and even other solution concepts like controller based policies. Even in the context of FOND problems there are open problems related to explaining properties of the policy as a whole. For example, are there methods from classical planning literature that could act as the basis for explaining why a given policy may be a strong-cyclic solution or a weak-cyclic one.

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
