# OpenReview forum: "Why Did You Do That? Generalizing Causal Link Explanations to Fully Observable Non-Deterministic Planning Problems"
_icaps-conference.org/ICAPS/2022/Workshop/XAIP — XAIP 2022_

### Official Review · Reviewer_cxL4 · 2022-04-17
**review for Why Did You Do That?**

**Rating:** 7
**Confidence:** 4

**Review:**

# paper summary
The paper is concerned with explaining plans (policies) in the context of non-determinism. If I got it right, authors assume well-justified plans (which implies that no single action can be removed as otherwise asking for their purpose might seem odd) and then use a chain of actions from the current action (the one asked about) to the goal as explanation. This idea is inspired by an approach for classical planning where a chain of causal links is followed until a goal. The authors show that the explanations they produce (Def. 2) are a generalization of those (based on causal links) by Seegebarth et al. by showing a correspondence between the two if restricted to determinism (Prop. 3).


# recommendation
I didn't find anything wrong, though I have to admit that I didn't check every single detail as I would have done for an A* conference submission. I read it carefully anyway and everything seems plausible and correct. Like the authors I am also not aware of any approaches for explaining plans in the context of planning with uncertainty. This work therefore appears to be the first, which increases its impact. Thus a clear accept from my side.


# detailed comments and suggestions
- I think you never really say how your explanation actually looks like. You start by saying that if a causal link exists, then we have justification for the action we want to get explained, but then you immediately go on saying that this alone doesn't even prove that the action is actually required, so go on stating more restricted criteria by Fink and Yang. But that leaves the reader questioning again what these causal links are used for. So, what is it? You should definitely spell that out. Are you following Seegebarth et al's approach and just trivially follow any chain of causal links to the goal and use the sequence of actions as explanation? (You need a1, because it's required for a2, which in turn .... etc.) Or do you want to answer the question *whether* a given action is required? In any case, I think this should be made clearer (before section 4 already)
- You make extensive use of causal links and even talk about causal threats -- both of which are usually used in partially ordered plans. In your work you seem however to restrict to totally ordered plans, so this seems something that should be mentioned. This does make a difference for example for plan optimization. To check whether a single action can be removed from a total-order plan (to find a well-justified plan) is clearly polynomial, but doing the same for a POCL plan is NP-complete (Olz & Bercher, ICAPS-2019). So it might make sense emphasizing that you use the concept of causal links, but still only use TO plans. (Not too important, though.)
- Isn't Definition 1 *exactly the same* as the corresponding definition of well-justified by Fink and Yang? If so, I wouldn't just call it "well-justified", but also add a reference to their paper and definition.
- In 4.1 you mention the all outcome determinization for the the fist time. Note that your paper might not only be read by FOND experts, but also by XAI folk with limited knowledge about FOND. I'd just quickly explain what it means (maybe in a footnote as most readers might know it)
- Since there seems to be a general correspondence between explanations in FOND planning and the deterministic setting (Prop. 3) it also seems equally important to use well-justified plans in the FOND setting just as it's in the deterministic setting as otherwise you would still compute an "explanation" (i.e., such a chain) although the action in question could just be removed. I don't think that this was stated (or emphasized) yet. Or you could offer what to do if it's not well-justified. A reasonable explanation to me would be: "I can't explain this as the action is actually not required!"
- Closely related to the last part: Why do you evaluate how often not well-justified actions occur? You are not evaluating your explanation generation approach that way but the quality of the solution generation approach. So this information seems a bit pointless for the specific purpose.
- (Just a comment.) I'm really surprised by the very low reported causal chain lengths. Provided that you don't only explain actions that are close to the goal this would imply that plans are really short (or at least their causal chains, as even a classical plan with n (arbitrary large) can have maximal chains of k (arbitrarily small), but that seems very unlikely). Did you restrict to overly short plans? Or are the problems just so very easy? Or did I get something wrong and you are actually shortening down the causal chains somehow?


# corrections
- your abstract is not an abstract. It's a short introduction. I would strongly advise to cut it down to at most half its length and move that 'story' into the introduction -- as this is what it is: an introduction, not an abstract.
- The paper has several missing or misplaced s:
 ~ "used explanations techniques" --> explanation
 ~ "and any state that satisfy" --> satisfies
 ~ "Each new action in the set Aπ correspond to⟨s,a⟩" --> corresponds
 ~ "in terms of both fluent set" --> sets
 ~ "of how the action contribute to" --> either actions or contributes
 ~ "The landmark set [...] correspond to" --> corresponds
 ~ "the set of valid explanation" --> explanations
 ~ "(a) the frequency with which well-justified action occurs" --> actions occur
 ~ "and five problems instance per domain" --> problem instances
 ~ "to generate such explanation." --> explanations
 ~ "even in the context of such generalized causal chain explanation" --> explanations
 ~ "every causal chain explanation correspond to" --> corresponds
- Some wrong commas:
 ~ "where, F is" --> no comma
 ~ "Figure 1, presents" --> no comma
 ~ "It’s also worth noting that, we are" --> no comma
 ~ "every causal chain explanation correspond to" --> no comma
 ~ "Table 1, provides an overview" --> no comma
 ~ "Possible information, here could" --> no comma
- "by a function π : S → A ∪ {a∅})" --> delete last parenthesis
- "a causal link is said to exist between two-step oi:ai and oj:aj if" --> "two-step" sounds *very* wrong. It should just be "two steps"
- "a clear notion of when action" --> "when an action"
- In the sentence having "one could informally say" in it, all starting apostrophe are wrong. You always wrote >>'a'<< and >>'s'<< (i.e., 99), but it should be >>`a'<< and >>`s'<< (i.e,. 69. Later it's correct)
- "One point to note here that is that to" --> eliminate first 'that'
- "using the D(Mπ) to ⟨s,a⟩" --> eliminate 'the' or add a description.
- "This brings us to the first proposition" --> missing dot at the end. It also appears a bit odd since no proposition follows.
- "we are guaranteed that G ⊆ L'" --> I think that mathcal(G) was never defined. (The goal description as well as the single goal predicate use the same symbol.)
- "then we have a correspond action" --> corresponding
- "Since the requirement ordering" --> required
- "such that all fi ∈ F is considered" --> are. Also, I think it's wrong to quantify over all fi from F. You must quantify over all fi for 1 <= i <= n. If you want to say that all fi are from F then you must say that Epsilon is an element from F^n.
- "(3.) holds because the causal links are preconditions" --> Causal links are not preconditions?
- "(b) time-taken to" --> time taken (time-taken isn't a word)
- Not really a "correction" as it's not "wrong", but I think it's much more 'natural' to read when fixed: Currently you only use one kind of citations: "[...]" no matter whether this citation is an object of your sentence or not. There are special commands for separating this. E.g., you could write:
 ~ "similar information was that of [Veloso, 1992]" --> of Veloso [1992]"
 ~ "was also used by [Chakraborti et al., 2019]" --> by Chakraborti et al. [2019]
 ~ "[Bryce et al., 2017] also looks" --> Bryce et al. [2017] also look at (note its plural now)
 (There are many more throughout the paper)
- Likewise you could compress down various identical citations; that might look more appealing. E.g., "[Sreedharan et al., 2021b; Sreedharan et al., 2019b]" --> [Sreedharan et al., 2021b; 2019b]
- Again purely appearance: You could take a look at the package (and its manual explaining design choices) 'booktabs', with which tables look 'cleaner' and just generally more appealing.
- Under the table, there's "[Vasileiou et al., ]", i.e., the bibtex is missing the year.
- "The thread of using abstraction to explain unsolvability in the context of FOND problems [Sreedharan et al., 2020a]." --> This is not a correct sentence, it's missing a verb. I also don't understand what it's supposed to say.
- "generalizing causal link to other planning formalisms" --> it's wrong language-wise (it should maybe say "causal links"), but it's also incorrect semantically because causal links are not a formalism, it's a data structure. Maybe you mean POCL planning formalism? But even that doesn't perfectly fit, so maybe just rephrase.
- "Now with the central property that we hope to explain in place" --> That sounds odd. It's your choice how you explain it, so there's nothing to hope for. Maybe you mean that you hope to *succeed* in explaining it in-place?
- "The former captured in terms of" --> "is captured"
- "both the actions capability" --> action's

---

### Official Review · Reviewer_iePF · 2022-04-27
**Good paper towards explaining non-deterministic planning problems, has minor issues**

**Rating:** 7
**Confidence:** 4

**Review:**

Summary:
This paper presents a method to extend causal link explanations to Fully Observable Non-Deterministic planning problems (FOND), to answer the question of "why is action 'a' required for the policy to reach the goal". In a non-deterministic setting, state transitions are uncertain, and the paper presents a way to deal with this by using concepts of justified actions and causal landmarks to explain why the action 'a' is required at state 's'. They propose a method to solve this problem via compilation to an unsolvability problem and provide empirical results on four domains, showing small runtimes in generating the explanation chains.

Review:
The paper has a key idea of an explainability paper: explaining why we need to do something. The methods presented in the paper appear to be able to solve the proposed problem of explaining state-action pairs in non-deterministic problems, where the solution is not a sequential plan but is rather a policy to be executed at each state, with uncertainty in the transitions. The work is a good extension of current ideas in XAIP and calls for extending XAIP to domains beyond classical planning, which is a necessary step forward.

Following are some comments about the content of the paper:
1. The paper presents a way to deal with explanations of why actions are necessary for a policy. It would be interesting to see whether the same methods can be applied to generating counter-factual explanations (i.e. why not 'b' instead of 'a').
2. The generation of new actions in section 4.1 could benefit from an example action.
3. Overall, the reviewer found section 4.1 difficult to follow. This is not a criticism of the contents, however it may be beneficial for readers to include brief high-level descriptions of the process and terms (e.g. determinization).
4. In the experimental results, including the number of states in each example or the typical length of a goal trace would help contextualize the chain length.  Currently, one can't tell if a chain length of 3 is good or not, as the entire trace from start to goal could have been 3 steps (meaning we explain the entire path), or it could have been 30 steps (meaning we only have to explain a subset of steps).

There are some minor typos throughout the paper.
e.g.:
 "every valid explanation corresponds to the fact that are part of.." -> "facts"
"every causal chain explanation correspond to an..." -> "corresponds"
"to be unsolvable as model will disallow any use..." -> "the model"

It is suggested that the authors proof-read and iron these out for the final submission.

---

### Meta-Review · Program_Chairs · 2022-04-30

**Recommendation:** Accept
**Confidence:** 5

**Metareview:**

The paper looks at explaining FOND policies through causal link explanations. It is one of the first papers to extend the scope of XAIP to planning problems with uncertainty. It will be a great addition to the workshop.

We hope the authors look carefully into the reviewers' comments and address them in the camera-ready submission. For example, it will help if the authors describe (or provide an example) of how the explanations will actually look like for a user. Finally, reviewer 2 pointed out fixes to language/typo errors in the paper; as they affect the legibility of the paper, please correct them for the final submission.

We are looking forward to an interesting and fruitful discussion at the workshop.

---

### Decision · Program_Chairs · 2022-04-30

Accept